# Early formative objective structured clinical examinations for students in the pre-clinical years of medical education: A non-randomized controlled prospective pilot study

**Naïm Ouldali**[1,2,3,4]*, **Enora Le Roux**[4], **Albert Faye**[1,3], **Claire Leblanc**[1],
**François Angoulvant**[1,5], **Diane Korb**[6,7], **Clémence Delcour**[6], **Caroline Caula**[8],
**Delphine Wohrer**[8], **Alexis Rybak**[3,8], **Manon Delafoy**[9], **Claire Carrié**[9], **Marion Strullu**[9],
**Mehdi Oualha**[10], **Romain Levy**[11], **Camille Mimoun**[12], **Lucie Griffon**[13],
**Alexandre Nuzzo**[14], **Clara Eyraud**[4], **Michael Levy**[15], **Pierre Ellul**[16,17]

1 Department of General Pediatrics, Pediatric Infectious Disease and Internal Medicine, Robert Debré University Hospital, Paris Cité University, Paris, France, 2 Infectious Diseases Division, CHU Sainte Justine - Montreal University, Montreal, Quebec, Canada, 3 ACTIV, Association Clinique et Thérapeutique Infantile du Val-de-Marne, Créteil, France, 4 Paris Cité University, INSERM UMR 1137, IAME (Infection, Antimicrobials, Modelling, Evolution), Paris, France, 5 INSERM, Cordeliers Research Center, UMRS 1138, Sorbonne Université, Université Paris Cité, Paris, France, 6 Department of Obstetrics and Gynecology, Robert Debré University Hospital, Paris University, Paris, France, 7 Paris University, Center for Epidemiology and Statistics Sorbonne Paris Cité (CRESS), Obstetrical Perinatal and Pediatric Epidemiology Research Team, EPOPé, INSERM, INRA, Paris, France, 8 Pediatric Emergency Department, Robert Debré University Hospital, Paris Cité University, Paris, France, 9 Department of Hematology, Robert Debré University Hospital, Paris Cité University, Paris, France, 10 Pediatric Intensive Care Unit, Necker University Hospital, Paris Cité University, Paris, France, 11 Immunology-Hematology and Rheumatology Department, Necker University Hospital, Paris Cité University, Paris, France, 12 Department of Gynecology and Obstetrics, Lariboisiere Hospital, Paris Cité University, Paris, France, 13 Pediatric Non-Invasive Ventilation and Sleep Unit, Necker Hospital, Paris Cité University, Paris, France, 14 Department of Gastroenterology, IBD and Intestinal Failure, Beaujon Hospital, Paris Cité University, Paris, France, 15 Pediatric Intensive Care Unit, Robert Debré University Hospital, Paris Cité University, Paris, France, 16 Child and Adolescent Psychiatry Department, Robert Debré University Hospital, Paris Cité University, Paris, France, 17 INSERM, Immunology-Immunopathology-Immunotherapy (i3), Sorbonne University, Paris, France

* naim.ouldali@aphp.fr

**Data Availability Statement:** The data that support the findings are owned by a third-party

## Abstract

### Background

The value of formative objective structured clinical examinations (OSCEs) during the pre-clinical years of medical education remains unclear. We aimed to assess the effectiveness of a formative OSCE program for medical students in their pre-clinical years on subsequent performance in summative OSCE.

### Methods

We conducted a non-randomized controlled prospective pilot study that included all medical students from the last year of the pre-clinical cycle of the Université Paris-Cité Medical School, France, in 2021. The intervention group received the formative OSCE program, which consisted of four OSCE sessions, followed by debriefing and feedback, whereas the

organization (the Paris University OSCE group) which were used under license for the current study, and so are not publicly available. Data are however available from the authors upon reasonable request to naim.ouldali@aphp.fr.

**Funding:** The authors received no specific funding for this work.

**Competing interests:** The authors have declared that no competing interests exist.

control group received the standard teaching program. The main objective of this formative OSCE program was to develop skills in taking a structured medical history and communication. All participants took a final summative OSCE. The primary endpoint was the summative OSCE mark in each group. A questionnaire was also administered to the intervention-group students to collect their feedback. A qualitative analysis, using a convenience sample, was conducted by gathering data pertaining to the process through on-site participative observation of the formative OSCE program.

## Results

Twenty students were included in the intervention group; 776 in the control group. We observed a significant improvement with each successive formative OSCE session in communication skills and in taking a structured medical history (p<0.0001 for both skills). Students from the intervention group performed better in a summative OSCE that assessed the structuring of a medical history (median mark 16/20, IQR [15; 17] versus 14/20, [13; 16], respectively, p = 0.012). Adjusted analyses gave similar results. The students from the intervention group reported a feeling of improved competence and a reduced level of stress at the time of the evaluation, supported by the qualitative data showing the benefits of the formative sessions.

## Conclusion

Our findings suggest that an early formative OSCE program is suitable for the pre-clinical years of medical education and is associated with improved student performance in domains targeted by the program.

## Background

The objective structured clinical examination (OSCE) [1], developed in the 1970's, is considered to be one of the most effective methods to assess the clinical competence of medical students in various disciplines [2, 3]. This approach can be used for formative or summative purposes [4]. Following its wide use in Canada and several European countries, summative OSCEs have been recently implemented in French medical schools [5, 6] and are part of the French Reform of the second cycle of medical studies (years 4 to 6 of medical studies) that will be implemented in 2022–2023 [7].

In France, as in many other countries, the first years of medical education are mainly dedicated to pre-clinical learning, without significative clinical or hospital exposure. Thus, OSCEs have initially been reserved for the clinical years of medical education that follow. However, such a pre-clinical/clinical dichotomy has recently been revised by several universities, paving the way for earlier implementation of OSCEs in the medical curriculum [8–10]. Among the fundamental early skills to be acquired, learning to take an extensive structured medical history is a key point during the pre-clinical years of medical education, and early OSCEs have the potential to improve this skill.

We conducted a literature review on the use of formative OSCEs in pre-clinical years of the medical formation. A recent study conducted in the United Kingdom suggested that early OSCEs integrated in the 2nd year of medical are both feasible and well perceived by students [8]. Similarly, early OSCEs targeting communication skills of first-year medical students in

Germany were also favorably received by students [11]. Similar findings have been reported in dental education [10, 12]. Most of these studies reported positive feedback from students concerning early OSCEs but almost none assessed the effectiveness of this tool on their future performance. Only one study described summative OSCE performance following formative OSCE and reported no significant improvement, except for the identical OSCE [9]. However, in this study, the skills targeted by the OSCE were mainly technical (such as taking a patient's blood pressure or subcutaneous injection) and no general skills taught in the pre-clinical years, such as semiology or taking a medical history, were assessed. Thus, the benefit of early OSCE in pre-clinical medical education is still unclear.

We aimed to assess the effectiveness of an early formative OSCE program dedicated to pre-clinical medical students on subsequent performance compared to students following a standard teaching program.

## Methods

### Study design

We conducted a non-randomized controlled prospective study within the Université Paris-Cité Medical School to evaluate the effectiveness of an early formative OSCE program for students in the preclinical years of their medical education on summative OSCE performance at the end of the last year of the pre-clinical cycle (i.e., third year of medical school in France) compared to students following the standard teaching program of the medical school without an OSCE. In addition, student feedback on the early formative OSCE program was also evaluated.

### Study population

All students in the last year of the preclinical cycle of medical studies of the Université Paris-Cité Medical School in 2020–2021 (1,002 students) were eligible for the study. The entire 2020–2021 class was divided into several groups by the medical school by alphabetical order to further distribute the students to different traineeships. The intervention group was one of such pre-established groups (20 students) and the rest constituted the control group. No student refused to participate in the study.

### Intervention

The formative OSCE program consisted of four sessions that were administered once every two weeks between April 1 and May 31, 2021. This program followed the key objectives of pre-clinical medical education in France, which focuses on taking an extensive structured medical history, as well as on developing good communication skills.

The 20 students from the intervention group were divided into four subgroups. For each, the OSCE sessions were organized as follows:

- Two teachers, both medical doctors, managed an OSCE station for a simulation lasting approximately 7 min, during which one teacher simulated the patient and the other was the examiner.

- Before each OSCE session, a short briefing was performed by one of the teachers detailing the objectives of the station.

- Following the OSCE, individual and collective debriefing sessions were conducted to collect students' feedback on positive points and difficulties.

The subject of each formative OSCE is detailed in S1 Appendix.

In both intervention and control groups, each student received the standard teaching program of semiology courses of the faculty, with a specific focus on medical history and had a daily 3-h observational traineeship in a hospital department for the last 12 weeks of the year.

## Primary endpoint and final evaluation

For all included students, a summative OSCE was performed at the end of the school year on May 2021, organized by the faculty. None of the teachers participating in the formative OSCE program were involved in the organization of the summative OSCE and were blinded to the summative OSCE subjects when conducting the formative program. The summative OSCE consisted of three stations (St1, 2, and 3), including one focusing on medical history, structuring of a medical history, and communication skills (St1, investigating the medical history of pelvic pain), and two others targeting technical skills or physical examination (St2 and St3, performing and analyzing an electrocardiogram and conducting the physical examination of a case of facial palsy, respectively).

The primary endpoint was the mark for the summative OSCE of the station targeting medical history (St1) in the intervention group compared to the control group. St2 and St3 served as the control outcome to explore the possibility of an extended benefit of formative OSCE to skills not specifically targeted by the program.

Secondary endpoints were the progression curve of the intervention group during the four OSCE training sessions, assessing communication skills and the ability to take an extensive structured medical history.

Finally, a standardized anonymized questionnaire was administered to the students of the intervention group to collect feedback concerning both the formative and summative OCSEs, including the level of induced stress, their personal feeling about the formative and summative OCSEs, self-confidence, and performance evaluated using a Likert scale for each item (S2 Appendix).

## Qualitative study

In addition, to complete the evaluation of this early OSCE training program, a qualitative study was conducted via the observation of five of the formative OSCE sessions (the four sessions of one subgroup and one session of another, convenience sampling). Marshall and Rossman define observation as "the systematic description of events, behaviors, and artifacts in the social setting chosen for study" [13]. This involves active looking, informal interviewing, and writing detailed field notes [14]. Two independent public health researchers trained in qualitative analysis (EL and CE) took extensive notes on students' behavior and attitudes during observation and did informal interviews with students before and after the sessions. The main purpose of the informal interviews' use here was to enable students to freely express what they think about the program. The "leader" interviewer was herself in the position of a student who can be considered a peer by the interviewees that is known to increased comfort of participants. [15] During the informal interviews the second researcher, senior, remained discreet. The questions used in informal interviews were not pre-written or structured and adapted to the spontaneous exchanges that took place between participants in the exercise. Informal interviews took place in the corridor before the start of the exercise, and at the end of the session in the room or the corridor lasting up to 10 minutes. The "leader" interviewer adopted active listening. The interviewers did not take notes during the exchanges so that these were as natural as possible, but reported the content of these, without interpretation, in raw data, immediately after the end of each session independently. The two researchers then compared their notes,

which allowed to reconstruct contents as completely as possible. Mixed methods were used in this study for triangulation purposes, in order to converge results from different methods with the goal of validating interpretations.

## Statistical analysis

The primary endpoint was analyzed using a two-tailed non-parametric Mann-Whitney test, as well as a univariate linear regression model. The same tests were used to analyze the progression curve during the four formative OSCEs.

Assuming a median mark of 12/20 in the control group for the summative OSCE, an improvement of 2 points in the intervention group and a standard deviation of 2, with a total of 16 students per group, the study was estimated to have 80% power to detect such a difference, assuming two-sided tests.

Multivariate linear regression models were then performed adjusted for age, repetition of the past year, and the number of students retaking exams to account for potential selection bias.

To deal with the difference in sample size between the intervention and the control group, we also conducted sub-group analyses by randomly selecting 80 students among the control group, thus providing a 1:4 ratio between the intervention and control groups respectively to reanalyze the main outcome. This process was performed 5 times, to explore whether the difference in sample size between the two groups could have influenced our findings.

The analyses were performed using R software (version 3.6.1), with a p value < 0.05 considered significant.

## Qualitative analysis

Inductive thematic analysis was used for qualitative analysis. Thus, we coded the data without trying to fit it into a preexisting coding frame, the analysis was data-driven [16]. After each observed session, the two qualitative researchers independently analyzed the digitally transcribed data collected from observation and informal interviews in tables build in a word processing software. Then, the two researchers compared and discussed their analysis and generated a summary with the consensual key themes that were identified, and the raw data associated. The result was then presented to the project team. A combination of the results of the qualitative and quantitative methods was used at the time the quantitative results were available. Data from qualitative analysis was linked with the quantitative results in an objective of illustration or enhancing interpretation.

# Results

## Implementation of the formative OSCE program

The preparation of the training program required 16 h to develop the exercise scenarios and evaluation grids and 4 h to organize the logistics carried out by four senior doctors (one senior per sub-group). Completion of the training program required the involvement of two senior doctors for 20 min per student, per session (for conducting a short briefing, the OSCE station, and individual debriefing) plus 15 min of collective debriefing per session. No expenditure was associated with implementation of the program.

## Student population and the formative OSCE program

Among the 1,002 medical students in the final year of the pre-clinical cycle in 2021 at the Université Paris-Cité Medical School, 206 did not take the final summative OSCE of the study and

were not included. Thus, 796 students were included in the study, of whom 20 were in the intervention group and 776 in the control group (flow chart, Fig 1). The baseline characteristics were similar between the two groups, including median age (21 years, IQR [21; 21] for both groups), the proportion of students having already repeated a year (7/20 [35%] in the intervention group vs 234/809 [29%] in the control group), and the median number retaking exams for the first semester (0, IQR [0; 1] for both groups).

Among the 20 students in the intervention group, 17 participated in the four formative OSCE sessions and three participated in only three.

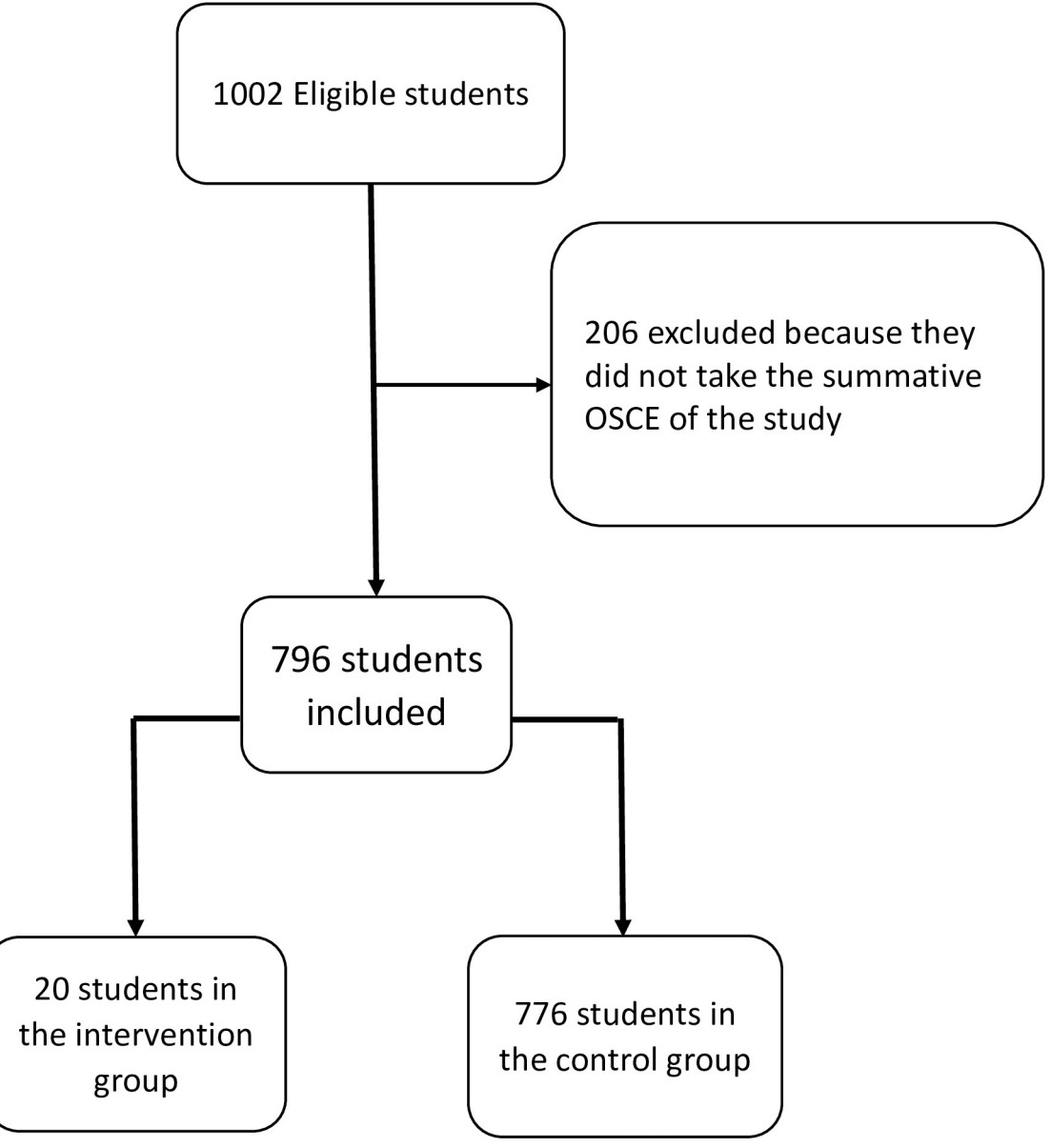

**Fig 1. Flow chart.** OSCE: objective structured clinical examination.

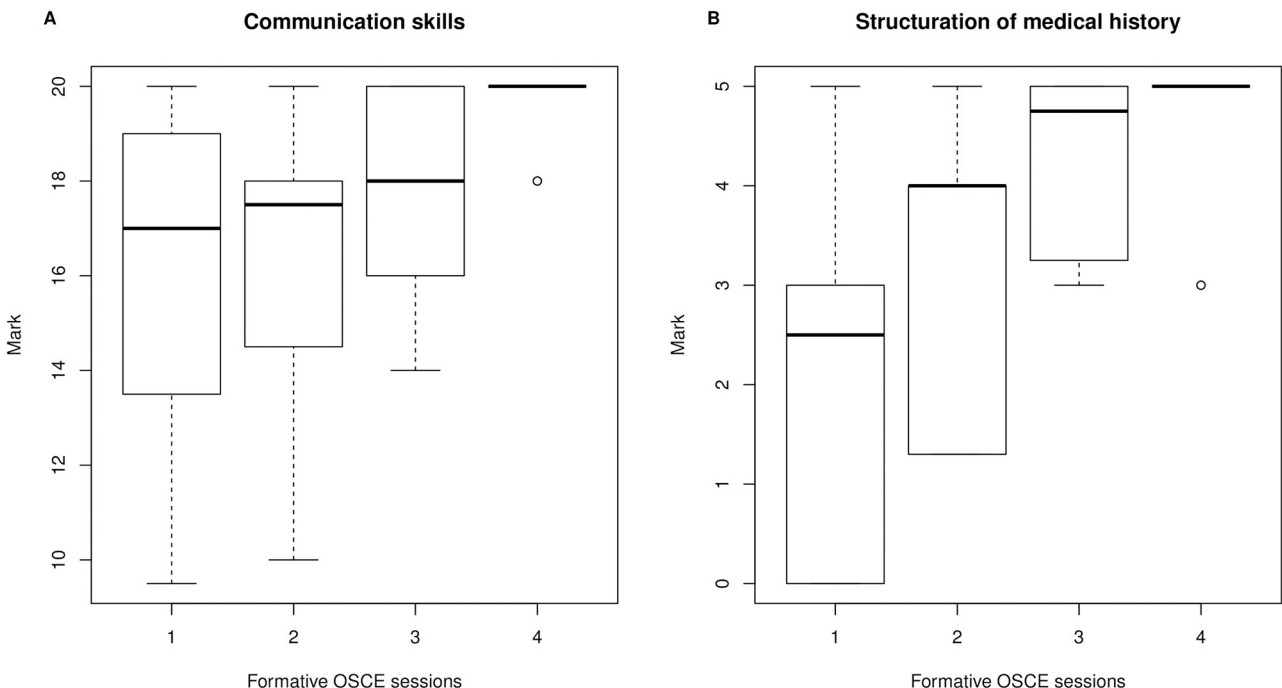

**Fig 2. Progression curve of students during the formative OSCE program (N = 20).** A) General communication skills. B) Structuring of a medical history. OSCE: objective structured clinical examination.

The progression curve during the formative OSCE program is presented in Fig 2. Both learning how to take a structured medical history and communication skills improved significantly during the intervention (p < 0.0001 for both skills).

## Performance in the subsequent summative OSCE

The median mark obtained for the early summative OSCE focusing on taking a structured medical history and communication skills was significantly higher for the intervention group than for the control group (median 16/20, IQR [15; 17] versus 14/20, [13; 16], respectively, p = 0.012, Table 1). Furthermore, the subcategory of the OSCE specifically assessing communication skills was also significantly higher for the intervention group (median mark 5/5, IQR [4; 5] vs 4/5 [3; 4] for the control group, p = 0.0002). Similar results were obtained when conducting multivariate linear regression adjusted for age, repetition of the past year, and the number

**Table 1. Performance of the intervention and control groups on a summative OSCE focusing on skills covered during the formative OSCE program (OSCE N˚1, structured medical history of a pelvic pain and communication skills), N = 796.**

|  | Intervention group (N = 20) | Control group (N = 776) | Univariate analysis | | Multivariate analysis* |
|---|---|---|---|---|---|
|  |  |  | Mann-Whitney test | Linear regression |  |
| Overall mark (/20), median [IQR] | 16.0 [14.8–17.0] | 14.0 [13.0–16.0] | P = 0.012 | P = 0.016 | P = 0.019 |
| Structuring of a medical history (/5), median [IQR] | 5.0 [4.0–5.0] | 4.0 [3.0; 4.0] | P = 0.0002 | P = 0.0005 | P = 0.0014 |

*Linear regression adjusted for age, repetition of the past year, and the number of students retaking exams. OSCE: objective structured clinical examination

**Table 2. Performance of the intervention and control groups on a summative OSCE focusing on skills not covered during the formative OSCE program (physical examination and technical skills), N = 796.**

| | Intervention group (N = 20) | Control group (N = 776) | Univariate analysis | | Multivariate analysis* |
|---|---|---|---|---|---|
| | | | Mann-Whitney test | Linear regression | |
| OSCE n˚2: Technical skills: performing and analyzing an electrocardiogram | 12.0 [8.8–16.0] | 14.0 [11.0–16.0] | P = 0.105 | P = 0.051 | P = 0.096 |
| OSCE n˚3: physical examination of a facial palsy | 14.5 [13.5–17.3] | 16.0 [14.0; 18.0] | P = 0.258 | P = 0.296 | P = 0.202 |

*: Linear regression adjusted for age, repetition of the past year, and the number of students retaking exams. OSCE: objective structured clinical examination

of students retaking exams (Table 1), or when conducting subgroup analyses with a 1:4 ratio between the intervention and the control group (S1 Fig).

However, the performance of the two groups did not significantly differ for the summative OSCE assessing skills not targeted by the formative OSCE program (physical examination or technical skill, without communication skills or medical history evaluation (Table 2)).

## Student feedback following the summative OSCE

Among the 20 students from the intervention group, 17 (85%) completed the final anonymized questionnaire to evaluate their experience. Regarding the structure of the program, 13/17 (76%) found that four formative OSCEs were sufficient, 2 (12%) suggested to reduce the number of formative OSCEs and 2 (12%) suggested to increase the number of sessions. The individual and collective debriefing were well perceived by 16/17 (94%) and 14/17 (82%) students, respectively. Regarding the usefulness of the program, 12/17 (71%) reported that the early formative OSCE program improved their performance. Regarding their personal experience, although 5/17 (29%) reported that the early OSCE program was stressful, 16/17 (94%) reported that it reduced their stress during the subsequent summative OSCE (S1 Fig).

## Qualitative analysis

The main themes that emerged from the observation analysis were related to stress, the ease and difficulties of the students to act as expected during the exercise, and the reported and observed benefits.

The qualitative data allowed interpretation of the quantitative results on the major results. First, concerning the improved performance of the intervention group during the exam, the observation data showed an improvement in performance from one formative session to another during the training, including increased fluidity of the student and higher satisfaction expressed by the evaluator during the end-of-session debriefing. A plateau effect in the acquisition of knowledge and skills was reported by several students and trainers after the fourth session. The students were not only trained by the exercise, i.e., how to properly conduct the exchange with the patient, but also by the feedback during the individual debriefing (what are the points to be addressed according to the scoring grid, how to ensure points, memory aid tips), which constituted a favorable context to pass the exam. On the other hand, qualitative data allowed the interpretation of non-positive results as well. One of the formative sessions (n ˚3/4) was dedicated to an exercise concerning the announcement of a diagnosis. This particularly unsettled the students. The ease, knowledge, and skills accumulated in previous sessions declined, the stress increased, and the evaluations were weak. Finally, concerning stress during the formative sessions, several students informally reported being stressed by being alone for

the first time (for the majority) in front of a patient (even simulated) and by being evaluated during the exercise, particularly by a referent teacher. "Score" and "validation" were major aspects in all debriefings. However, the repetition of the exercise, the better knowledge of assessment processes, and the debriefings were perceived as benefits. Debriefings were times of benevolent reassurance for the student and involved (no standardization) anchoring of the exercise in the professional future, a systematic review of the evaluation grid, and discussion with the evaluator. Overall, we observed good acceptability by the students, as well as by the doctors involved in the exercise, which created a favorable atmosphere for learning.

## Discussion

In this study, early formative OSCE dedicated to pre-clinical medical students was associated with improved subsequent performance compared to the standard teaching program. Our results highlight the positive impact of early formative OSCE on performance, as there was improvement only for the topics covered by the formative stations in the summative OSCE. Although the positive effect of formative OSCEs on summative outcomes is well described [17], few studies have explored their relevance in the pre-clinical cycle. Previous studies found that early OSCEs are feasible, reliable, and valid in the context of pre-clinical teaching [10, 12]. Only one study assessed the impact of early OSCEs targeting technical skills on subsequent OSCEs and found that the impact was restricted to identical technical summative OSCEs [9]. Overall, these findings suggest that it would be more beneficial to implement early formative OSCEs on general topics rather than technical issues [9]. Indeed, the knowledge expected in the pre-clinical training phases often involve broader skills, such as communication, conducting a structured interview, and learning semiology, which can be secondarily generalized, unlike technical skills. As several medical schools started to introduce clinical work earlier in the medical curriculum [8–10], this may also ease OSCE implementation at the early stage of the medical teaching program.

In our study, we found that the positive impact of early formative OSCEs did not extend to areas not covered by the formative OSCE program. It is possible that a summative OSCE assessing skills not covered by the training may have a destabilizing effect, in which the student trained on a specific aspect and facing an unknown exercise does not know how to use his/her knowledge and skills better than an untrained student. This is corroborated by our qualitative study, showing that the formative session unexpectedly requiring new skills from students was particularly disrupting. This may require diversification of the formative sessions, with the limitation of having to increase committed human resources.

The number of formative OSCE sessions appears to be an important consideration for the optimization of resource allocation to such a program. In our study, we found a positive effect of four formative sessions on the summative OSCE. A recent study assessing the impact of a single formative OSCE did not find any effect on subsequent performance [18]. There is currently no evidence for a minimum number of early formative OSCE sessions to improve summative OSCE performance. Indeed, performance in summative OSCE may not be directly related to direct teaching but to other factors, such as improved self-confidence and better management of exam stress, which was corroborated by qualitative data, suggesting a plateau effect at the fourth formative OSCE session.

Indeed, we confirmed that students positively accepted early formative OSCE with a feeling of increased competence and decreased stress. We believe that these results are partially related to the design of our study. Before each session, a short briefing was given to explain the skills assessed during the session and it has been previously shown that formative OSCEs are most effective when students understand the end goal and perceive the outcome as being aligned

with the stated expectations of the summative OSCE [19]. Thus, we also set up short individual and collective debriefings after the sessions. It is now clear that students appreciated and benefited more from the formative OSCE experience when the educational objective and assessment criteria were clearly explained [20]. Our students' responses to the questionnaire and the qualitative data support these results and highlight the need for systematic feedback delivered using a benevolent approach. Indeed, we assume that a major role of early formative OSCEs is to explain the process of OSCEs to students and to provide them feedback on their performance, thus increasing performance in latter summative OSCEs.

We show this program to be feasible and acceptable to all parties. It was carried out with existing resources as an integral part of the activity of university physicians. For its transferability to be ensured in other contexts, a balance must be found between the available time of university physicians, available resources, the number of involved students, and the optimal number of sessions to be planned.

This study had several limitations. First, our intervention group could be considered as being small (20 students). However, it was calculated *a priori*, to have sufficient power to detect a potential difference between groups, and subgroup analyses with a 1:4 ratio between the intervention and control groups found unchanged results. Second, a non-randomized study design was used that can potentially induce selection bias. However, the two groups were comparable to each other and multivariate linear regression adjusted for the students' baseline characteristics was performed to further limit the risk of bias, with the results remaining unchanged. In this context, our findings should be confirmed by further controlled studies, with larger sample size, more summative OSCE stations and other tests to allow triangulation of data, as suggested by recent recommendations on performance assessment. [21, 22] Third, the progression curve observed in this study should be interpreted in the context of a possible progress testing, that should also be assessed in further studies. [23] Fourth, formative OSCE sessions were performed by medical personal known by students rather than standardized simulated patients. This could lead to a loss of credibility during the OSCE. Further studies are required to compare the impact of different formative programs using standardized simulated patients. Finally, given the short duration of our study, the long-term effect of early formative OSCEs on further medical skills, particularly at the bedside, could not be evaluated.

In conclusion, this study suggests that an early formative OSCE program dedicated to the pre-clinical years of medical education may be associated with improved subsequent performance. These findings favor the early implementation of formative OSCEs to develop essential pre-clinical skills, such as learning to communicate with the patient, conducting a structured medical interview, and learning the basics of semiology. Further studies are needed to replicate these results and explore the value of early formative OSCE assessments in other core competency areas in medical studies.

## Supporting information

**S1 Appendix. Details of the four formative OSCE stations.**
(DOCX)

**S2 Appendix. Questionnaire administered to students from the intervention group.**
(DOCX)

**S1 Fig. Results of the questionnaire administered to intervention group students, N = 17.**
(DOCX)

## Acknowledgments

We thank Victoire de Lastours for her help and comments. We thank Damien Livoreil and the Paris Univerisity administrative team for their logistical support.

## Author Contributions

**Conceptualization:** Naïm Ouldali, Enora Le Roux, Albert Faye, François Angoulvant, Alexandre Nuzzo, Michael Levy, Pierre Ellul.

**Data curation:** Naïm Ouldali, Enora Le Roux, Claire Leblanc, Diane Korb, Clémence Delcour, Caroline Caula, Delphine Wohrer, Alexis Rybak, Manon Delafoy, Claire Carrié, Marion Strullu, Clara Eyraud, Pierre Ellul.

**Formal analysis:** Naïm Ouldali, Clara Eyraud, Pierre Ellul.

**Methodology:** Naïm Ouldali, Enora Le Roux, Albert Faye, Claire Leblanc, François Angoulvant, Diane Korb, Clémence Delcour, Caroline Caula, Manon Delafoy, Mehdi Oualha, Alexandre Nuzzo, Clara Eyraud, Michael Levy, Pierre Ellul.

**Project administration:** François Angoulvant.

**Resources:** Claire Leblanc, Alexandre Nuzzo, Michael Levy.

**Supervision:** Enora Le Roux, François Angoulvant, Mehdi Oualha.

**Validation:** Naïm Ouldali, Enora Le Roux, Albert Faye, Claire Leblanc, Clémence Delcour, Caroline Caula, Delphine Wohrer, Alexis Rybak, Manon Delafoy, Claire Carrié, Marion Strullu, Mehdi Oualha, Romain Levy, Camille Mimoun, Lucie Griffon, Alexandre Nuzzo, Clara Eyraud, Michael Levy, Pierre Ellul.

**Writing – original draft:** Naïm Ouldali, Pierre Ellul.

**Writing – review & editing:** Naïm Ouldali, Enora Le Roux, Albert Faye, Claire Leblanc, François Angoulvant, Diane Korb, Clémence Delcour, Caroline Caula, Delphine Wohrer, Alexis Rybak, Manon Delafoy, Claire Carrié, Marion Strullu, Mehdi Oualha, Romain Levy, Camille Mimoun, Lucie Griffon, Alexandre Nuzzo, Clara Eyraud, Michael Levy, Pierre Ellul.

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
