## [Decision Letter · Decision Letter 0]

19 Sep 2022

PONE-D-22-23055Early formative objective structured clinical examinations for students in the pre-clinical years of medical education: a non-randomized controlled prospective pilot study.PLOS ONE

Dear Dr. Ouldali,

Thank you for submitting your manuscript to PLOS ONE. After careful consideration, we feel that it has merit but does not fully meet PLOS ONE’s publication criteria as it currently stands. Therefore, we invite you to submit a revised version of the manuscript that addresses the points raised during the review process.

I am happy to have been involved in the evaluation process of this article. It is important that article authors are educators working in clinical sciences. Because this type of research is generally in the field of interest of educators working in Medical Education departments. I think that there are aspects that are not clear enough in the article in terms of the fact that the research topic is remarkable and reader expectations about the subject. Some of these are the extreme difference between the number of students in the case and control groups, and the inadequacy of explanation for qualitative research, which is especially mentioned in the reviewer suggestions. The fact that the research was conducted with a mixed methodology necessitates detailed explanations about the methodology. Particular attention should be paid to clarity and detailed expression as a criterion of validity and reliability in the qualitative dimension. For the topic covered in the article, I would expect different training methods, such as training with simulated patients, to at least be included in the discussion. Therefore, I think it would be appropriate to correct the article.

We look forward to receiving your revised manuscript.

Kind regards,

Ayse Hilal Bati, Associate Professor

Academic Editor

PLOS ONE

Additional Editor Comments :

Dear Author/s,

I am happy to have been involved in the evaluation process of this article. It is important that article authors are educators working in clinical sciences. Because this type of research is generally in the field of interest of educators working in Medical Education departments. I think that there are aspects that are not clear enough in the article in terms of the fact that the research topic is remarkable and reader expectations about the subject. Some of these are the extreme difference between the number of students in the case and control groups, and the inadequacy of explanation for qualitative research, which is especially mentioned in the reviewer suggestions. The fact that the research was conducted with a mixed methodology necessitates detailed explanations about the methodology. Particular attention should be paid to clarity and detailed expression as a criterion of validity and reliability in the qualitative dimension.

For the topic covered in the article, I would expect different training methods, such as training with simulated patients, to at least be included in the discussion.

Therefore, I think it would be appropriate to correct the article.

Reviewers' comments:

Reviewer's Responses to Questions

**Comments to the Author**

1. Is the manuscript technically sound, and do the data support the conclusions?

Reviewer #1: Yes

Reviewer #2: Yes

2. Has the statistical analysis been performed appropriately and rigorously? 

Reviewer #1: I Don't Know

Reviewer #2: Yes

3. Have the authors made all data underlying the findings in their manuscript fully available?

Reviewer #1: No

Reviewer #2: Yes

4. Is the manuscript presented in an intelligible fashion and written in standard English?

Reviewer #1: Yes

Reviewer #2: Yes

5. Review Comments to the Author

Reviewer #1: This simple study takes the hypothesis that a formative assessment task prior to a summative assessment task is likely to result in increased performance in that summative task. Authors use some nice techniques to show that the effect is limited to tasks specifically included in the formative task and for the intervention group. They claim the study is sufficiently powered despite the large discrepancy in case vs control group sizes. Whilst there have been similar studies in the literature over the years, the authors are keen to make a distinction between this study in the pre-clinical years, albeit focusing on a clinical skill (history-taking), and other studies based in clinical years. I feel this is stretched in this day, since many medical programs have long since mixed clinical work with clinical science learning. Nonetheless there is a contribution to the literature here.

1. The study presents the results of original research.

Yes

2. Results reported have not been published elsewhere, to the best of my knowledge.

Yes

3. Experiments, statistics, and other analyses are performed to a high technical standard and are described in sufficient detail.

I am not an expert in statistical analysis but the analysis appears cogent from my perspective.

4. Conclusions are presented in an appropriate fashion and are supported by the data.

Yes

5. The article is presented in an intelligible fashion and is written in standard English.

Yes - there are some minor typos but nothing more e.g. personal used instead of personnel.

6. The research meets all applicable standards for the ethics of experimentation and research integrity.

Yes

7. The article adheres to appropriate reporting guidelines and community standards for data availability.

The authors state they will provide data but seeing that I am not a statistical expert that would not help in interpretation.

Reviewer #2: Thank-you for submitting this manuscript which is very well written and reports an interesting study.

My only concern regards the qualitative component of the work, which I do not think is presented in adequate detail.

This needs to be addressed with regard to the following issues:

- It is stated on page 12, lines 238-240 that the main themes that emerged from the qualitative analysis were related to....."

Were these the actual main themes? If so this should be made clear. Were there any sub-themes?

- More detail is needed regarding data collection. How many interviews were conducted with how many students? What questions were asked? In what format were the interview? Where were they conducted? Were they recorded?

- Likewise how the data analyses was conducted need further explanation. Were the interviews audio recorded and transcribed, for example? If so, by whom? How were the themes decided on?

- It would also be useful for some quotes from the students to be included to illustrate/ support the main themes.

If these issues cannot be addressed, I recommend removing this entire section and reference to it from the paper.

6. PLOS authors have the option to publish the peer review history of their article (what does this mean?). If published, this will include your full peer review and any attached files.

Reviewer #1: No

Reviewer #2: No

---

## [Author Response · Author response to Decision Letter 0]

11 Oct 2022

Naïm Ouldali, MD, PhD

General Pediatrics, infectious diseases and internal medicine department

Robert Debré University Hospital 

48 Bd Sérurier, 75019 Paris, France. 

Email: naim.ouldali@aphp.fr. 

Phone: 0033140032048

 Ayse Hilal Bati 

 Academic Editor

 PLOS ONE

 Paris, October 11th, 2022

Re-submission of manuscript PONE-D-22-23055 entitled: “Early formative objective structured clinical examinations for students in the pre-clinical years of medical education: a non-randomized controlled prospective pilot study” 

 Dear Editor,

We would like to thank you for giving us the opportunity to submit the revised manuscript entitled “Early formative objective structured clinical examinations for students in the pre-clinical years of medical education: a non-randomized controlled prospective pilot study”. We took into account the reviewers’ comments and suggestions and modified the manuscript accordingly. 

You will find below the point-by-point responses outlining the modifications made to the manuscript and the additional analyses performed. As requested, we have submitted the revised manuscript incorporating the changes below, together with a marked-up copy of the changes made from the previous article as tracked changes. 

We hope that this revised and improved version of our manuscript will be suitable for publication in PLOS ONE. 

 Sincerely,

 Naïm Ouldali, MD, PhD.

 

PONE-D-22-23055

Early formative objective structured clinical examinations for students in the pre-clinical years of medical education: a non-randomized controlled prospective pilot study.

PLOS ONE

Dear Dr. Ouldali,

Thank you for submitting your manuscript to PLOS ONE. After careful consideration, we feel that it has merit but does not fully meet PLOS ONE’s publication criteria as it currently stands. Therefore, we invite you to submit a revised version of the manuscript that addresses the points raised during the review process.

I am happy to have been involved in the evaluation process of this article. It is important that article authors are educators working in clinical sciences. Because this type of research is generally in the field of interest of educators working in Medical Education departments. 

I think that there are aspects that are not clear enough in the article in terms of the fact that the research topic is remarkable and reader expectations about the subject. Some of these are the extreme difference between the number of students in the case and control groups, and the inadequacy of explanation for qualitative research, which is especially mentioned in the reviewer suggestions. 

• First, we would like to thank the editor and the reviewers for their careful review and their important suggestions provided to improve the manuscript. 

• We fully agree that the difference between the number of students in the two groups is an issue. As suggested and to address this, we performed an additional analysis by randomly selecting 80 students among the control group, thus providing a 1:4 ratio between the intervention and control groups, and reanalyzed the main outcome among this sub-group. 

We performed this process 5 times, and in all cases, the results remained very similar. 

• This additional analysis is now included in the revised manuscript (appendix 3). The following sentences have been added: 

o Method section line 182: “To deal with the difference in sample size between the intervention and the control group, we also conducted sub-group analyses by randomly selecting 80 students among the control group, thus providing a 1:4 ratio between the intervention and control groups to reanalyze the main outcome. This process was performed 5 times, to explore whether the difference in sample size between the two groups could have influenced our findings.”

o Results section, line 234: “Similar results were obtained when conducting multivariate linear regression adjusted for age, repetition of the past year, and the number of students retaking exams (Table 1), or when conducting subgroup analyses with a 1:4 ratio between the intervention and the control group (appendix 3).”

o Discussion section, line 335: “This study had several limitations. First, our intervention group could be considered as being small (20 students). However, it was calculated a priori, to have sufficient power to detect a potential difference between groups, and subgroup analyses with a 1:4 ratio between the intervention and control groups found unchanged results.”

The fact that the research was conducted with a mixed methodology necessitates detailed explanations about the methodology. Particular attention should be paid to clarity and detailed expression as a criterion of validity and reliability in the qualitative dimension. 

• We thank the editor and the reviewer #2 for this important comment and suggestion. As requested, we clarified the method section reporting the qualitative study, to avoid misunderstanding regarding the qualitative method used (participative observation of the formative OSCE program and not formal interviews). As suggested by the reviewer #2 and the editor, the following changes were made: 

o Method section, line 160: “In addition, to complete the evaluation of this early OSCE training program, a qualitative study was conducted via the observation of five of the formative OSCE sessions (the four sessions of one subgroup and one session of another, convenience sampling). Marshall and Rossman defined observation as "the systematic description of events, behaviors, and artifacts in the social setting chosen for study" [13]. This involves active looking, informal interviewing, and writing detailed field notes [14]. Two independent public health researchers trained in qualitative analysis (EL and CE) took extensive notes on students’ behavior and attitudes during observation and did informal interviews with students before and after the sessions. Mixed methods were used in this study for triangulation purposes, in order to converge results from different methods with the goal of validating interpretations.”

o Method section, line 191: “Inductive thematic analysis was used for qualitative analysis. Thus, we coded the data without trying to fit it into a preexisting coding frame. The analysis was data-driven [15].”

o Results section, line 253: “The main themes that emerged from the observation analysis were related to stress, the ease and difficulties of the students to act as expected during the exercise, and the reported and observed benefits.”

For the topic covered in the article, I would expect different training methods, such as training with simulated patients, to at least be included in the discussion. 

• We fully agree with the editor. As suggested, to better highlight this point, the following changes were made: 

o Discussion section line 342: “Third, formative OSCE sessions were performed by medical personal known by students rather than standardized simulated patients. This could lead to a loss of credibility during the OSCE. However, a recent study showed that known medical examiners were perceived to be more credible than standardized patients [17]. One explanation relies on social identity theory, which suggests that students identified known medical personal as their ‘in-group’ versus the ‘‘out-group’’ for standardized patients, influencing the perception of each group [18]. Further studies are required to compare the impact of different formative programs using standardized simulated patients.”

Therefore, I think it would be appropriate to correct the article.

• We thank the editor for giving us the opportunity to correct the article, and hope that the changes provided are suitable. We are looking forward to a continued discussion for any remaining issues or additional questions.

We look forward to receiving your revised manuscript.

Kind regards,

Ayse Hilal Bati, Associate Professor

Academic Editor

PLOS ONE

• Done

• We thank the editor for this remark. Indeed, the data that support the findings are owned by a third-party organization (the Paris University OSCE group), but can be available upon request. As suggested, we have updated this paragraph in the manuscript accordingly. 

• Done

Additional Editor Comments :

Dear Author/s,

I am happy to have been involved in the evaluation process of this article. It is important that article authors are educators working in clinical sciences. Because this type of research is generally in the field of interest of educators working in Medical Education departments. I think that there are aspects that are not clear enough in the article in terms of the fact that the research topic is remarkable and reader expectations about the subject. Some of these are the extreme difference between the number of students in the case and control groups, and the inadequacy of explanation for qualitative research, which is especially mentioned in the reviewer suggestions. The fact that the research was conducted with a mixed methodology necessitates detailed explanations about the methodology. Particular attention should be paid to clarity and detailed expression as a criterion of validity and reliability in the qualitative dimension.

For the topic covered in the article, I would expect different training methods, such as training with simulated patients, to at least be included in the discussion.

Therefore, I think it would be appropriate to correct the article.

Reviewers' comments:

Reviewer's Responses to Questions

Comments to the Author

1. Is the manuscript technically sound, and do the data support the conclusions?

Reviewer #1: Yes

Reviewer #2: Yes

2. Has the statistical analysis been performed appropriately and rigorously? 

Reviewer #1: I Don't Know

Reviewer #2: Yes

3. Have the authors made all data underlying the findings in their manuscript fully available?

Reviewer #1: No

Reviewer #2: Yes

4. Is the manuscript presented in an intelligible fashion and written in standard English?

Reviewer #1: Yes

Reviewer #2: Yes

5. Review Comments to the Author

Reviewer #1: This simple study takes the hypothesis that a formative assessment task prior to a summative assessment task is likely to result in increased performance in that summative task. Authors use some nice techniques to show that the effect is limited to tasks specifically included in the formative task and for the intervention group. They claim the study is sufficiently powered despite the large discrepancy in case vs control group sizes. Whilst there have been similar studies in the literature over the years, the authors are keen to make a distinction between this study in the pre-clinical years, albeit focusing on a clinical skill (history-taking), and other studies based in clinical years. I feel this is stretched in this day, since many medical programs have long since mixed clinical work with clinical science learning. Nonetheless there is a contribution to the literature here.

• We thank the reviewer for this comment and these suggestions. 

• We fully agree that the difference between the number of students in the two groups is an issue. As suggested, to address this, we performed an additional analysis by randomly selecting 80 students among the control group, thus providing a 1:4 ratio between the intervention and control groups, and reanalyzed the main outcome among this sub-group. 

We performed this process 5 times, and in all cases, the results remained very similar. 

• This additional analysis is now included in the revised manuscript (appendix 3). The following sentences have been added: 

o Method section line 182: “To deal with the difference in sample size between the intervention and the control group, we also conducted sub-group analyses by randomly selecting 80 students among the control group, thus providing a 1:4 ratio between the intervention and control groups to reanalyze the main outcome. This process was performed 5 times, to explore whether the difference in sample size between the two groups could have influenced our findings.”

o Results section, line 234: “Similar results were obtained when conducting multivariate linear regression adjusted for age, repetition of the past year, and the number of students retaking exams (Table 1), or when conducting subgroup analyses with a 1:4 ratio between the intervention and the control group (appendix 3).”

o Discussion section, line 335: “This study had several limitations. First, our intervention group could be considered as being small (20 students). However, it was calculated a priori, to have sufficient power to detect a potential difference between groups. Subgroup analyses with a 1:4 ratio between the intervention and control groups found unchanged results.”

• Regarding the distinction between preclinical and clinical years, we fully agree with the reviewer that some medical teaching programs now include clinical work early in the medical program. To better discuss this point, the following changes were made: 

o Discussion section line 297: “As several universities started to introduce clinical work earlier in the medical curriculum [8–10], this may also ease OSCE implementation at the early stage of the medical teaching program."

1. The study presents the results of original research.

Yes

2. Results reported have not been published elsewhere, to the best of my knowledge.

Yes

3. Experiments, statistics, and other analyses are performed to a high technical standard and are described in sufficient detail.

I am not an expert in statistical analysis but the analysis appears cogent from my perspective.

4. Conclusions are presented in an appropriate fashion and are supported by the data.

Yes

5. The article is presented in an intelligible fashion and is written in standard English.

Yes - there are some minor typos but nothing more e.g. personal used instead of personnel.

6. The research meets all applicable standards for the ethics of experimentation and research integrity.

Yes

7. The article adheres to appropriate reporting guidelines and community standards for data availability.

The authors state they will provide data but seeing that I am not a statistical expert that would not help in interpretation.

• We thank the reviewer for these positive comments. 

 

Reviewer #2: Thank-you for submitting this manuscript which is very well written and reports an interesting study.

My only concern regards the qualitative component of the work, which I do not think is presented in adequate detail.

This needs to be addressed with regard to the following issues:

- 1. It is stated on page 12, lines 238-240 that the main themes that emerged from the qualitative analysis were related to....."

Were these the actual main themes? If so this should be made clear. Were there any sub-themes?

- 2. More detail is needed regarding data collection. How many interviews were conducted with how many students? What questions were asked? In what format were the interview? Where were they conducted? Were they recorded?

-3. Likewise how the data analyses was conducted need further explanation. Were the interviews audio recorded and transcribed, for example? If so, by whom? How were the themes decided on?

-4. It would also be useful for some quotes from the students to be included to illustrate/ support the main themes.

If these issues cannot be addressed, I recommend removing this entire section and reference to it from the paper.

• We thank the reviewer for these comments and questions which prompted us to clarify the use of qualitative methods in our article. We did not carry out formal interviews, the qualitative data were collected from observations and informal exchanges that took place before and after the formative OSCE sessions. 

• Participant observation has been a hallmark of both anthropological and sociological studies for many years. In recent years, the field of education has seen an increase in the number of qualitative studies that include participant observation as a way to collect information (Kawulich, 2005). Marshall and Rossman defined observation as "the systematic description of events, behaviors, and artifacts in the social setting chosen for study" (Marshall, 1989). Observations enable the researcher to describe existing situations using the five senses, providing a "written photograph" of the situation under study (Erlandson, 1993). It involves active looking, informal interviewing, and detailed field notes writing (DeWalt, 2002). DeWalt and DeWalt (2002) suggested that participant observation should be used as a way to increase the validity of the study, as observations may help the researcher have a better understanding of the context and phenomenon under study. Validity is stronger with the use of additional strategies used with observation, such as quantitative methods – in our study, questionnaires (DeWalt, 2002). All these reasons led us to choose this qualitative research method to ensure the description of our educational innovation and better interpret the results of the quantitative analysis. 

• When writing up one's ethnographic observations, it is advised to follow the lead of Spradley & McCurdy (1972) and find a cultural scene, spend time with the informants, asking questions and clarifying answers, analyze the material, pulling together the themes into a well-organized story. We therefore carried out an inductive thematic analysis and the process of coding the data was done without trying to fit it into a preexisting coding frame. Thus, we had no pre-identified themes. The analysis was data-driven (Braun & Clarke, 2006). Although extensive notes were taken during the observations, the material analyzed was less rich than in other qualitative methods such as structured interviews. Thus, the elaborate thematic tree was simple, without sub-theme.

• As suggested by the reviewer, to better describe the qualitative study and avoid misunderstanding, the following changes were made:

o Method section, line 160: “In addition, to complete the evaluation of this early OSCE training program, a qualitative study was conducted via the observation of five of the formative OSCE sessions (the four sessions of one subgroup and one session of another, convenience sampling). Marshall and Rossman defined observation as "the systematic description of events, behaviors, and artifacts in the social setting chosen for study" [13]. It involves active looking, informal interviewing, and writing detailed field notes [14]. Two independent public health researchers trained in qualitative analysis (EL and CE) took extensive notes on students’ behavior and attitudes during observation and did informal interviews with students before and after the sessions. Mixed methods were used in this study for triangulation purposes, which aim to converge results from different methods with the goal of validating interpretations.”

o Method section, line 191: “Inductive thematic analysis was used for qualitative analysis, we therefore code the data without trying to fit it into a pre-specified coding frame, the analysis was data-driven [15].”

o Results section, line 253: “The main themes that emerged from the observation analysis were related to stress, the ease and difficulties of the students to act as expected during the exercise, and the reported and observed benefits.”

• Specific answer to the reviewer remarks: 

- 1. It is stated on page 12, lines 238-240 that the main themes that emerged from the qualitative analysis were related to....."

Were these the actual main themes? If so this should be made clear. Were there any sub-themes?

• Yes, these were the main themes, without sub-themes, as explained below.

- 2. More detail is needed regarding data collection. How many interviews were conducted with how many students? What questions were asked? In what format were the interview? Where were they conducted? Were they recorded?

• No interview was conducted, as explained above. The qualitative study was conducted via the observation of five of the formative OSCE sessions (method section, line 161). 

-3. Likewise how the data analyses was conducted need further explanation. Were the interviews audio recorded and transcribed, for example? If so, by whom? How were the themes decided on?

• No interview was recorded. The observation of the formative OSCE sessions was performed by Drs Enora Le Roux and Clara Eyraud. As frequently used for this qualitative method, the analysis was data-driven, without pre-identified themes (Braun & Clarke, 2006). 

-4. It would also be useful for some quotes from the students to be included to illustrate/ support the main themes.

• As no interview was conducted, no quotes were included in the manuscript. 

• We hope that the changes made to the manuscript allow a better understanding of the qualitative method used in this study. We are looking forward to a continued discussion for any remaining issues.

References

Kawulich, Barbara. Participant Observation as a Data Collection Method, Forum: Qualitative Social Research 6 (2), Art. 43 – May 200. DOI: https://doi.org/10.17169/fqs-6.2.466

Marshall, Catherine & Rossman, Gretchen B. (1989). Designing qualitative research. Newbury Park, CA: Sage. ISBN : 0803931581, 9780803931589

Erlandson, David A.; Harris, Edward L.; Skipper, Barbara L. & Allen, Steve D. (1993). Doing naturalistic inquiry: a guide to methods. Newbury Park, CA: Sage. ISBN: 9780803949386

DeWalt, Kathleen M. & DeWalt, Billie R. (2002). Participant observation: a guide for fieldworkers. Walnut Creek, CA: AltaMira Press. ISBN : 0759119279

Spradley, James P. & McCurdy, David W. (1972). The Cultural Experience. Chicago: Science Research Associates. ISBN : 1577663640

Braun V., Clarke V. (2006). Using thematic analysis in psychology. Qualitative Research in Psychology, 3, 77–101. doi:10.1191/1478088706qp063oa

---

## [Decision Letter · Decision Letter 1]

1 Dec 2022

PONE-D-22-23055R1

Early formative objective structured clinical examinations for students in the pre-clinical years of medical education: a non-randomized controlled prospective pilot study.

PLOS ONE

Dear Dr. Ouldali,

Thank you for submitting your manuscript to PLOS ONE. After careful consideration, we have decided that your manuscript does not meet our criteria for publication and must therefore be rejected.

I am sorry that we cannot be more positive on this occasion, but hope that you appreciate the reasons for this decision.

Kind regards,

Ayse Hilal Bati, Associate Professor

Academic Editor

PLOS ONE

Reviewers' comments:

Reviewer's Responses to Questions

**Comments to the Author**

1. If the authors have adequately addressed your comments raised in a previous round of review and you feel that this manuscript is now acceptable for publication, you may indicate that here to bypass the “Comments to the Author” section, enter your conflict of interest statement in the “Confidential to Editor” section, and submit your "Accept" recommendation.

Reviewer #1: All comments have been addressed

Reviewer #3: (No Response)

2. Is the manuscript technically sound, and do the data support the conclusions?

Reviewer #1: Yes

Reviewer #3: Partly

3. Has the statistical analysis been performed appropriately and rigorously? 

Reviewer #1: Yes

Reviewer #3: Yes

4. Have the authors made all data underlying the findings in their manuscript fully available?

Reviewer #1: Yes

Reviewer #3: No

5. Is the manuscript presented in an intelligible fashion and written in standard English?

Reviewer #1: Yes

Reviewer #3: Yes

6. Review Comments to the Author

Reviewer #1: (No Response)

Reviewer #3: Thank you for the opportunity to review the paper. The authors are to be applauded for developing a formative OSCE. This is a pilot study of an intervention for 20 students comparing OSCE test results for the intervention group compared to the other students (n=776).

Although this may be considered novel locally, there is a vast literature demonstrating that formative OSCEs predict future performance. Thus, the findings from this study do not add substantially to the current body of literature. (see relationship to other variables in the Boursicot article from 2020; Martin 2002)

In order to advance the literature, it is important to consider modern validity theory (Boursicot 2020; Cook 2006)

For example, content validity in examinations (and the need for blueprinting as knowledge and skills are content specific) have been known for several years. (Dory 2010) Hence, a 3 station OSCE would not be considered a valid endpoint.

The “progression curve”, as noted, should be discussed in the context of progress testing, which also has a significant body of literature. One example is the article by Pugh 2016.

The qualitative analysis is lacking in detail. Most qualitative researchers would not consider “informal interviews” a sufficient description in the methods.

Overall, the team would do well to add a medical educator to their research team and consider reaching out to authors on references provided (who are approachable and have decades of research experience in performance- based exams, including OSCEs).

Boursicot K, Kemp S, Wilkinson T. et al. Performance assessment: Consensus statement and recommendations from the 2020 Ottawa Conference. Medical Teacher 2021, 43:1, 58-67

Martin I, Jolly B. Predictive validity and estimated cut score of an objective structured clinical examination (OSCE) used as an assessment of clinical skills at the end of the first clinical year. Med Ed 2002, 36(5), p. 418.

Cook D, Beckman T. Current Concepts in Validity and Reliability for Psychometric Instruments: Theory and Application, The American Journal of Medicine 2006, 199; 166.

Dory V, Gagnon R, Charlin B. Is case-specificity content-specificity? An analysis of data from extended-matching questions. Adv Health Sci Educ Theory Pract; 55-63.

Pugh D, Bhanji F, Cole G, Dupre J, Hatala R, Humphrey-Murto S, Touchie C, Wood TJ. Do OSCE Progress Test Scores predict performance in a national high-stakes examination? Medical Education. 2016; 50(3):351-8. doi: 10.1111/medu.12942.

7. PLOS authors have the option to publish the peer review history of their article (what does this mean?). If published, this will include your full peer review and any attached files.

Reviewer #1: No

Reviewer #3: No

- - - - -

---

## [Author Response · Author response to Decision Letter 1]

17 May 2023

PONE-D-22-23055

Early formative objective structured clinical examinations for students in the pre-clinical years of medical education: a non-randomized controlled prospective pilot study.

PLOS ONE

1. If the authors have adequately addressed your comments raised in a previous round of review and you feel that this manuscript is now acceptable for publication, you may indicate that here to bypass the “Comments to the Author” section, enter your conflict of interest statement in the “Confidential to Editor” section, and submit your "Accept" recommendation.

Reviewer #1: All comments have been addressed

Reviewer #3: (No Response)

2. Is the manuscript technically sound, and do the data support the conclusions?

Reviewer #1: Yes

Reviewer #3: Partly

3. Has the statistical analysis been performed appropriately and rigorously?

Reviewer #1: Yes

Reviewer #3: Yes

4. Have the authors made all data underlying the findings in their manuscript fully available?

Reviewer #1: Yes

Reviewer #3: No

5. Is the manuscript presented in an intelligible fashion and written in standard English?

Reviewer #1: Yes

Reviewer #3: Yes

6. Review Comments to the Author

Please use the space provided to explain your answers to the questions above. You may also include additional comments for the author, including concerns about dual publication, research ethics, or publication ethics. (Please upload your review as an attachment if it exceeds 20,000 characters).

Reviewer #1: (No Response)

Reviewer #3: Thank you for the opportunity to review the paper. The authors are to be applauded for developing a formative OSCE. This is a pilot study of an intervention for 20 students comparing OSCE test results for the intervention group compared to the other students (n=776).

Although this may be considered novel locally, there is a vast literature demonstrating that formative OSCEs predict future performance. Thus, the findings from this study do not add substantially to the current body of literature. (see relationship to other variables in the Boursicot article from 2020; Martin 2002)

- Response: We thank the reviewer for this general comment. We fully agree with the reviewer that there is a vast literature that already demonstrated the benefit of formative OSCE. However, the scope of our article is somehow different as we aimed to assess the benefit of formative OSCE in the specific context of the pre-clinical years of the medical formation. Indeed, as OSCE focuses on clinical competences, it is in most cases used after the 3rd year of medical studies, where the clinical exposure is substantial. Thus, its benefit in pre-clinical years remains poorly studied.

- As suggested by the reviewer, we conducted a new literature review on the specific question of formative OSCE effectiveness in pre-clinical years of the medical formation. We only identified one article, that was previously cited in the introduction and discussion section of our manuscript (reference #9). This article assessed the benefit of OSCE targeting technical skills (such as taking a patient’s blood pressure or subcutaneous injection) and the general skills taught in the pre-clinical years, such as semiology or taking a medical history, were not covered. 

- As suggested to clarify this important point, the following changes were made to the manuscript, with references indicated by the reviewer added to the manuscript (Boursicot et al is now reference 22 of the article): 

o Abstract section, line 47: “We aimed to assess the effectiveness of a formative OSCE program for medical students in their pre-clinical years on subsequent performance in summative OSCE.”

o Abstract section, line 70: “. Our findings suggest that an early formative OSCE program is suitable for the pre-clinical years of medical education and is associated with improved student performance in domains targeted by the program.”

o Introduction section, line 88: “We conducted a literature review on the use of formative OSCEs in pre-clinical years of the medical formation.”

o Discussion section, line 301: “In this study, early formative OSCE dedicated to pre-clinical medical students was associated with improved subsequent performance compared to the standard teaching program.”

o Limit section, line 360: “In this context, our findings should be confirmed by further controlled studies, with larger sample size, more summative OSCE stations and other tests to allow triangulation of data, as suggested by recent recommendations on performance assessment.[21, 22]”

- We hope that these explanations and the changes provided to the manuscript will allow clarifying the specific objective of this study, and we are ready to make other changes if the reviewer or the editor require it. 

In order to advance the literature, it is important to consider modern validity theory (Boursicot 2020; Cook 2006). 

For example, content validity in examinations (and the need for blueprinting as knowledge and skills are content specific) have been known for several years. (Dory 2010) Hence, a 3 station OSCE would not be considered a valid endpoint.

- Response: We thank the reviewer for this comment. We fully agree that the modern validity theory is important to consider. As a pilot study, the aim of this work was not to definitely establish the benefit of formative OSCE in pre-clinical years of medical education, but rather to provide first results regarding the effectiveness of this formative OSCE program, that should be further validated by other studies. 

- To better highlight the potential benefit of this formative OSCE program, we also performed an evaluation with 2 “control outcomes”. These controls consisted in summative OSCE stations that did not target topics covered by the formative OSCE program. Interestingly, no differences were found between the two groups for these two stations, suggesting that the improvement following formative OSCE is specific to the topics covered by the formative stations, thereby reinforcing the association between the formative OSCE program and outcomes. 

- As suggested to better highlight this limitation, the following sentences were changed, and references indicated by the reviewer were added: 

o Limit section, line 360: “Second, a non-randomized study design was used that can potentially induce selection bias. However, the two groups were comparable to each other and multivariate linear regression adjusted for the students’ baseline characteristics was performed to further limit the risk of bias, with the results remaining unchanged. In this context, our findings should be confirmed by further controlled studies, with larger sample size, more summative OSCE stations and other tests to allow triangulation of data, as suggested by recent recommendations on performance assessment.[21, 22]”

The “progression curve”, as noted, should be discussed in the context of progress testing, which also has a significant body of literature. One example is the article by Pugh 2016.

- Response: We thank the reviewer for this remark. As suggested to highlight this point, the following sentences were added, along with reference indicated by the reviewer (reference 23): 

o Discussion section, line 361: “Third, the progression curve observed in this study should be interpreted in the context of a possible progress testing, that should also be assessed in further studies.[23]”

The qualitative analysis is lacking in detail. Most qualitative researchers would not consider “informal interviews” a sufficient description in the methods.

- Response: We thank the reviewer for this comment which allowed us to clarify this qualitative part. We have added details to the description of the qualitative analysis method and justified more clearly the use of informal interviews. 

- The main purpose of the informal interviews’ use here was to enable students to freely express what they thought about the program. These informal interviews create a greater ease of communication and often produce more naturalistic data (Swain). The interviewers, as recommended (Zhang), were skilled in stimulating respondents to open up about what they thought. In our study it was facilitated by the fact that the “leader” interviewer was herself in the position of a student who can be considered a peer by the interviewees that is known to increased comfort of participants (Devota). During the informal interviews the second researcher, senior, remained discreet.

- The questions used in informal interviews were not pre-written or structured and adapted to the spontaneous exchanges that took place between participants in the exercise. Informal interviews took place in the corridor within 15 to 10 minutes before the opening of the room before the start of the exercise, and at the end of the session in the room or the corridor lasting up to 10 minutes. The students then joined their clinical services where they carried out their internships. 

- The leader interviewer adopted active listening meaning she encouraged extended responses by showing an interest in the topics being discussed and asks questions that encourage the speaker to reveal more (Pearson). The interviewers did not take notes during the exchanges so that these were as natural as possible, but reported the content of these, without interpretation, in raw data, immediately after the end of each session independently. The two researchers then compared their notes, which allowed to reconstruct as many contents as possible.

References: 

Zhang, G., Stufflebeam, D. L. (2017). The CIPP Evaluation Model: How to Evaluate for Improvement and Accountability. United Kingdom: Guilford Publications.

Devotta, K., Woodhall-Melnik, J., Pedersen, C., Wendaferew, A., Dowbor, T. P., Guilcher, S. J., Hamilton-Wright, S., Ferentzy, P., Hwang, S. W., & Matheson, F. I. (2016). Enriching qualitative research by engaging peer interviewers: a case study. Qualitative Research, 16(6), 661–680. https://doi.org/10.1177/1468794115626244

Swain, J., & King, B. (2022). Using Informal Conversations in Qualitative Research. International Journal of Qualitative Methods, 21. https://doi.org/10.1177/16094069221085056

Pearson, J., Nelson, P., Titsworth, S., and Harter, L. (2006). Human Communication (2nd ed.). Boston, MA: McGraw Hill.

- As suggested by the reviewer, to better describe the qualitative method, the following changes were made:

o Method section, line 161: 

“Qualitative study

In addition, to complete the evaluation of this early OSCE training program, a qualitative study was conducted via the observation of five of the formative OSCE sessions (the four sessions of one subgroup and one session of another, convenience sampling). Marshall and Rossman define observation as "the systematic description of events, behaviors, and artifacts in the social setting chosen for study" [13]. This involves active looking, informal interviewing, and writing detailed field notes [14]. Two independent public health researchers trained in qualitative analysis (EL and CE) took extensive notes on students’ behavior and attitudes during observation and did informal interviews with students before and after the sessions. The main purpose of the informal interviews’ use here was to enable students to freely express what they think about the program. The “leader” interviewer was herself in the position of a student who can be considered a peer by the interviewees that is known to increased comfort of participants.[15] During the informal interviews the second researcher, senior, remained discreet. The questions used in informal interviews were not pre-written or structured and adapted to the spontaneous exchanges that took place between participants in the exercise. Informal interviews took place in the corridor before the start of the exercise, and at the end of the session in the room or the corridor lasting up to 10 minutes. The “leader” interviewer adopted active listening. The interviewers did not take notes during the exchanges so that these were as natural as possible, but reported the content of these, without interpretation, in raw data, immediately after the end of each session independently. The two researchers then compared their notes, which allowed to reconstruct contents as completely as possible. Mixed methods were used in this study for triangulation purposes, in order to converge results from different methods with the goal of validating interpretations. 

[…]

Qualitative analysis 

Inductive thematic analysis was used for qualitative analysis. Thus, we coded the data without trying to fit it into a preexisting coding frame. The analysis was data-driven [16]. After each observed session, the two qualitative researchers independently analyzed the digitally transcribed data collected from observation and informal interviews in tables build in a word processing software. Then, the two researchers compared and discussed their observations analysis and generated a summary with the consensual key themes that were identified and the raw data associated. Data from informal interviews were added. The result was then presented to the project team. A combination of the results of the qualitative and quantitative methods was used at the time the quantitative results were available interpretated by the entire study steering group. Data from qualitative analysis was linked with the quantitative results in an objective of illustration or enhancing interpretation.”

Overall, the team would do well to add a medical educator to their research team and consider reaching out to authors on references provided (who are approachable and have decades of research experience in performance- based exams, including OSCEs).

- Response: We thank the reviewer for this suggestion. This work has been supervised by Prof Albert Faye, who has worked for 15 years as medical educator within Paris Diderot then Paris Cité University, and is the vice-president of the education council of the Paris Cité University medical school. He is an expert of this field, as testified by numerous co-authored publications in international peer-reviewed journals (some of the most recent related to OSCE: 

o doi: 10.1080/0142159X.2023.2198665

o doi: 10.1186/s12909-022-03919-1.

o doi: 10.1371/journal.pone.0245439.

o doi: 10.1371/journal.pone.0238542.

Dr Naïm Ouldali, Dr Enora Leroux, Dr Michael Levy and Prof François Angoulvant, co-authors of this study, are also medical educators in Paris Cité University.

7. PLOS authors have the option to publish the peer review history of their article. If published, this will include your full peer review and any attached files.

Do you want your identity to be public for this peer review? For information about this choice, including consent withdrawal, please see our Privacy Policy.

Reviewer #1: No

Reviewer #3: No

 

Naïm Ouldali, MD, PhD

General Pediatrics, infectious diseases and internal medicine department

Robert Debré University Hospital 

48 Bd Sérurier, 75019 Paris, France. 

Email: naim.ouldali@aphp.fr. 

Phone: 0033140032048

Ayse Hilal Bati 

Academic Editor

PLOS ONE

Paris, October 11th, 2022

Re-submission of manuscript PONE-D-22-23055 entitled: “Early formative objective structured clinical examinations for students in the pre-clinical years of medical education: a non-randomized controlled prospective pilot study” 

Dear Editor,

We would like to thank you for giving us the opportunity to submit the revised manuscript entitled “Early formative objective structured clinical examinations for students in the pre-clinical years of medical education: a non-randomized controlled prospective pilot study”. We took into account the reviewers’ comments and suggestions and modified the manuscript accordingly. 

You will find below the point-by-point responses outlining the modifications made to the manuscript and the additional analyses performed. As requested, we have submitted the revised manuscript incorporating the changes below, together with a marked-up copy of the changes made from the previous article as tracked changes. 

We hope that this revised and improved version of our manuscript will be suitable for publication in PLOS ONE. 

 Sincerely,

Naïm Ouldali, MD, PhD.

 

PONE-D-22-23055

Early formative objective structured clinical examinations for students in the pre-clinical years of medical education: a non-randomized controlled prospective pilot study.

PLOS ONE

Dear Dr. Ouldali,

Thank you for submitting your manuscript to PLOS ONE. After careful consideration, we feel that it has merit but does not fully meet PLOS ONE’s publication criteria as it currently stands. Therefore, we invite you to submit a revised version of the manuscript that addresses the points raised during the review process.

I am happy to have been involved in the evaluation process of this article. It is important that article authors are educators working in clinical sciences. Because this type of research is generally in the field of interest of educators working in Medical Education departments. 

I think that there are aspects that are not clear enough in the article in terms of the fact that the research topic is remarkable and reader expectations about the subject. Some of these are the extreme difference between the number of students in the case and control groups, and the inadequacy of explanation for qualitative research, which is especially mentioned in the reviewer suggestions. 

• First, we would like to thank the editor and the reviewers for their careful review and their important suggestions provided to improve the manuscript. 

• We fully agree that the difference between the number of students in the two groups is an issue. As suggested and to address this, we performed an additional analysis by randomly selecting 80 students among the control group, thus providing a 1:4 ratio between the intervention and control groups, and reanalyzed the main outcome among this sub-group. 

We performed this process 5 times, and in all cases, the results remained very similar. 

• This additional analysis is now included in the revised manuscript (appendix 3). The following sentences have been added: 

o Method section line 182: “To deal with the difference in sample size between the intervention and the control group, we also conducted sub-group analyses by randomly selecting 80 students among the control group, thus providing a 1:4 ratio between the intervention and control groups to reanalyze the main outcome. This process was performed 5 times, to explore whether the difference in sample size between the two groups could have influenced our findings.”

o Results section, line 234: “Similar results were obtained when conducting multivariate linear regression adjusted for age, repetition of the past year, and the number of students retaking exams (Table 1), or when conducting subgroup analyses with a 1:4 ratio between the intervention and the control group (appendix 3).”

o Discussion section, line 335: “This study had several limitations. First, our intervention group could be considered as being small (20 students). However, it was calculated a priori, to have sufficient power to detect a potential difference between groups, and subgroup analyses with a 1:4 ratio between the intervention and control groups found unchanged results.”

The fact that the research was conducted with a mixed methodology necessitates detailed explanations about the methodology. Particular attention should be paid to clarity and detailed expression as a criterion of validity and reliability in the qualitative dimension. 

• We thank the editor and the reviewer #2 for this important comment and suggestion. As requested, we clarified the method section reporting the qualitative study, to avoid misunderstanding regarding the qualitative method used (participative observation of the formative OSCE program and not formal interviews). As suggested by the reviewer #2 and the editor, the following changes were made: 

o Method section, line 160: “In addition, to complete the evaluation of this early OSCE training program, a qualitative study was conducted via the observation of five of the formative OSCE sessions (the four sessions of one subgroup and one session of another, convenience sampling). Marshall and Rossman defined observation as "the systematic description of events, behaviors, and artifacts in the social setting chosen for study" [13]. This involves active looking, informal interviewing, and writing detailed field notes [14]. Two independent public health researchers trained in qualitative analysis (EL and CE) took extensive notes on students’ behavior and attitudes during observation and did informal interviews with students before and after the sessions. Mixed methods were used in this study for triangulation purposes, in order to converge results from different methods with the goal of validating interpretations.”

o Method section, line 191: “Inductive thematic analysis was used for qualitative analysis. Thus, we coded the data without trying to fit it into a preexisting coding frame. The analysis was data-driven [15].”

o Results section, line 253: “The main themes that emerged from the observation analysis were related to stress, the ease and difficulties of the students to act as expected during the exercise, and the reported and observed benefits.”

For the topic covered in the article, I would expect different training methods, such as training with simulated patients, to at least be included in the discussion. 

• We fully agree with the editor. As suggested, to better highlight this point, the following changes were made: 

o Discussion section line 342: “Third, formative OSCE sessions were performed by medical personal known by students rather than standardized simulated patients. This could lead to a loss of credibility during the OSCE. Further studies are required to compare the impact of different formative programs using standardized simulated patients.”

Therefore, I think it would be appropriate to correct the article.

• We thank the editor for giving us the opportunity to correct the article, and hope that the changes provided are suitable. We are looking forward to a continued discussion for any remaining issues or additional questions.

We look forward to receiving your revised manuscript.

Kind regards,

Ayse Hilal Bati, Associate Professor

Academic Editor

PLOS ONE

• Done

• We thank the editor for this remark. Indeed, the data that support the findings are owned by a third-party organization (the Paris University OSCE group), but can be available upon request. As suggested, we have updated this paragraph in the manuscript accordingly. 

• Done

Additional Editor Comments :

Dear Author/s,

I am happy to have been involved in the evaluation process of this article. It is important that article authors are educators working in clinical sciences. Because this type of research is generally in the field of interest of educators working in Medical Education departments. I think that there are aspects that are not clear enough in the article in terms of the fact that the research topic is remarkable and reader expectations about the subject. Some of these are the extreme difference between the number of students in the case and control groups, and the inadequacy of explanation for qualitative research, which is especially mentioned in the reviewer suggestions. The fact that the research was conducted with a mixed methodology necessitates detailed explanations about the methodology. Particular attention should be paid to clarity and detailed expression as a criterion of validity and reliability in the qualitative dimension.

For the topic covered in the article, I would expect different training methods, such as training with simulated patients, to at least be included in the discussion.

Therefore, I think it would be appropriate to correct the article.

Reviewers' comments:

Reviewer's Responses to Questions

Comments to the Author

1. Is the manuscript technically sound, and do the data support the conclusions?

Reviewer #1: Yes

Reviewer #2: Yes

2. Has the statistical analysis been performed appropriately and rigorously? 

Reviewer #1: I Don't Know

Reviewer #2: Yes

3. Have the authors made all data underlying the findings in their manuscript fully available?

Reviewer #1: No

Reviewer #2: Yes

4. Is the manuscript presented in an intelligible fashion and written in standard English?

Reviewer #1: Yes

Reviewer #2: Yes

5. Review Comments to the Author

Reviewer #1: This simple study takes the hypothesis that a formative assessment task prior to a summative assessment task is likely to result in increased performance in that summative task. Authors use some nice techniques to show that the effect is limited to tasks specifically included in the formative task and for the intervention group. They claim the study is sufficiently powered despite the large discrepancy in case vs control group sizes. Whilst there have been similar studies in the literature over the years, the authors are keen to make a distinction between this study in the pre-clinical years, albeit focusing on a clinical skill (history-taking), and other studies based in clinical years. I feel this is stretched in this day, since many medical programs have long since mixed clinical work with clinical science learning. Nonetheless there is a contribution to the literature here.

• We thank the reviewer for this comment and these suggestions. 

• We fully agree that the difference between the number of students in the two groups is an issue. As suggested, to address this, we performed an additional analysis by randomly selecting 80 students among the control group, thus providing a 1:4 ratio between the intervention and control groups, and reanalyzed the main outcome among this sub-group. 

We performed this process 5 times, and in all cases, the results remained very similar. 

• This additional analysis is now included in the revised manuscript (appendix 3). The following sentences have been added: 

o Method section line 182: “To deal with the difference in sample size between the intervention and the control group, we also conducted sub-group analyses by randomly selecting 80 students among the control group, thus providing a 1:4 ratio between the intervention and control groups to reanalyze the main outcome. This process was performed 5 times, to explore whether the difference in sample size between the two groups could have influenced our findings.”

o Results section, line 234: “Similar results were obtained when conducting multivariate linear regression adjusted for age, repetition of the past year, and the number of students retaking exams (Table 1), or when conducting subgroup analyses with a 1:4 ratio between the intervention and the control group (appendix 3).”

o Discussion section, line 335: “This study had several limitations. First, our intervention group could be considered as being small (20 students). However, it was calculated a priori, to have sufficient power to detect a potential difference between groups. Subgroup analyses with a 1:4 ratio between the intervention and control groups found unchanged results.”

• Regarding the distinction between preclinical and clinical years, we fully agree with the reviewer that some medical teaching programs now include clinical work early in the medical program. To better discuss this point, the following changes were made: 

o Discussion section line 297: “As several medical schools started to introduce clinical work earlier in the medical curriculum [8–10], this may also ease OSCE implementation at the early stage of the medical teaching program."

1. The study presents the results of original research.

Yes

2. Results reported have not been published elsewhere, to the best of my knowledge.

Yes

3. Experiments, statistics, and other analyses are performed to a high technical standard and are described in sufficient detail.

I am not an expert in statistical analysis but the analysis appears cogent from my perspective.

4. Conclusions are presented in an appropriate fashion and are supported by the data.

Yes

5. The article is presented in an intelligible fashion and is written in standard English.

Yes - there are some minor typos but nothing more e.g. personal used instead of personnel.

6. The research meets all applicable standards for the ethics of experimentation and research integrity.

Yes

7. The article adheres to appropriate reporting guidelines and community standards for data availability.

The authors state they will provide data but seeing that I am not a statistical expert that would not help in interpretation.

• We thank the reviewer for these positive comments. 

 

Reviewer #2: Thank-you for submitting this manuscript which is very well written and reports an interesting study.

My only concern regards the qualitative component of the work, which I do not think is presented in adequate detail.

This needs to be addressed with regard to the following issues:

- 1. It is stated on page 12, lines 238-240 that the main themes that emerged from the qualitative analysis were related to....."

Were these the actual main themes? If so this should be made clear. Were there any sub-themes?

- 2. More detail is needed regarding data collection. How many interviews were conducted with how many students? What questions were asked? In what format were the interview? Where were they conducted? Were they recorded?

-3. Likewise how the data analyses was conducted need further explanation. Were the interviews audio recorded and transcribed, for example? If so, by whom? How were the themes decided on?

-4. It would also be useful for some quotes from the students to be included to illustrate/ support the main themes.

If these issues cannot be addressed, I recommend removing this entire section and reference to it from the paper.

• We thank the reviewer for these comments and questions which prompted us to clarify the use of qualitative methods in our article. We did not carry out formal interviews, the qualitative data were collected from observations and informal exchanges that took place before and after the formative OSCE sessions. 

• Participant observation has been a hallmark of both anthropological and sociological studies for many years. In recent years, the field of education has seen an increase in the number of qualitative studies that include participant observation as a way to collect information (Kawulich, 2005). Marshall and Rossman defined observation as "the systematic description of events, behaviors, and artifacts in the social setting chosen for study" (Marshall, 1989). Observations enable the researcher to describe existing situations using the five senses, providing a "written photograph" of the situation under study (Erlandson, 1993). It involves active looking, informal interviewing, and detailed field notes writing (DeWalt, 2002). DeWalt and DeWalt (2002) suggested that participant observation should be used as a way to increase the validity of the study, as observations may help the researcher have a better understanding of the context and phenomenon under study. Validity is stronger with the use of additional strategies used with observation, such as quantitative methods – in our study, questionnaires (DeWalt, 2002). All these reasons led us to choose this qualitative research method to ensure the description of our educational innovation and better interpret the results of the quantitative analysis. 

• When writing up one's ethnographic observations, it is advised to follow the lead of Spradley & McCurdy (1972) and find a cultural scene, spend time with the informants, asking questions and clarifying answers, analyze the material, pulling together the themes into a well-organized story. We therefore carried out an inductive thematic analysis and the process of coding the data was done without trying to fit it into a preexisting coding frame. Thus, we had no pre-identified themes. The analysis was data-driven (Braun & Clarke, 2006). Although extensive notes were taken during the observations, the material analyzed was less rich than in other qualitative methods such as structured interviews. Thus, the elaborate thematic tree was simple, without sub-theme.

• As suggested by the reviewer, to better describe the qualitative study and avoid misunderstanding, the following changes were made:

o Method section, line 160: “In addition, to complete the evaluation of this early OSCE training program, a qualitative study was conducted via the observation of five of the formative OSCE sessions (the four sessions of one subgroup and one session of another, convenience sampling). Marshall and Rossman defined observation as "the systematic description of events, behaviors, and artifacts in the social setting chosen for study" [13]. It involves active looking, informal interviewing, and writing detailed field notes [14]. Two independent public health researchers trained in qualitative analysis (EL and CE) took extensive notes on students’ behavior and attitudes during observation and did informal interviews with students before and after the sessions. Mixed methods were used in this study for triangulation purposes, which aim to converge results from different methods with the goal of validating interpretations.”

o Method section, line 191: “Inductive thematic analysis was used for qualitative analysis, we therefore code the data without trying to fit it into a pre-specified coding frame, the analysis was data-driven [15].”

o Results section, line 253: “The main themes that emerged from the observation analysis were related to stress, the ease and difficulties of the students to act as expected during the exercise, and the reported and observed benefits.”

• Specific answer to the reviewer remarks: 

- 1. It is stated on page 12, lines 238-240 that the main themes that emerged from the qualitative analysis were related to....."

Were these the actual main themes? If so this should be made clear. Were there any sub-themes?

• Yes, these were the main themes, without sub-themes, as explained below.

- 2. More detail is needed regarding data collection. How many interviews were conducted with how many students? What questions were asked? In what format were the interview? Where were they conducted? Were they recorded?

• No interview was conducted, as explained above. The qualitative study was conducted via the observation of five of the formative OSCE sessions (method section, line 161). 

-3. Likewise how the data analyses was conducted need further explanation. Were the interviews audio recorded and transcribed, for example? If so, by whom? How were the themes decided on?

• No interview was recorded. The observation of the formative OSCE sessions was performed by Drs Enora Le Roux and Clara Eyraud. As frequently used for this qualitative method, the analysis was data-driven, without pre-identified themes (Braun & Clarke, 2006). 

-4. It would also be useful for some quotes from the students to be included to illustrate/ support the main themes.

• As no interview was conducted, no quotes were included in the manuscript. 

• We hope that the changes made to the manuscript allow a better understanding of the qualitative method used in this study. We are looking forward to a continued discussion for any remaining issues.

References

Kawulich, Barbara. Participant Observation as a Data Collection Method, Forum: Qualitative Social Research 6 (2), Art. 43 – May 200. DOI: https://doi.org/10.17169/fqs-6.2.466

Marshall, Catherine & Rossman, Gretchen B. (1989). Designing qualitative research. Newbury Park, CA: Sage. ISBN : 0803931581, 9780803931589

Erlandson, David A.; Harris, Edward L.; Skipper, Barbara L. & Allen, Steve D. (1993). Doing naturalistic inquiry: a guide to methods. Newbury Park, CA: Sage. ISBN: 9780803949386

DeWalt, Kathleen M. & DeWalt, Billie R. (2002). Participant observation: a guide for fieldworkers. Walnut Creek, CA: AltaMira Press. ISBN : 0759119279

Spradley, James P. & McCurdy, David W. (1972). The Cultural Experience. Chicago: Science Research Associates. ISBN : 1577663640

Braun V., Clarke V. (2006). Using thematic analysis in psychology. Qualitative Research in Psychology, 3, 77–101. doi:10.1191/1478088706qp063oa

---

## [Decision Letter · Decision Letter 2]

25 Oct 2023

Early formative objective structured clinical examinations for students in the pre-clinical years of medical education: a non-randomized controlled prospective pilot study.

PONE-D-22-23055R2

Dear Dr. Ouldali,

We’re pleased to inform you that your manuscript has been judged scientifically suitable for publication and will be formally accepted for publication once it meets all outstanding technical requirements.

Kind regards,

Ayse Hilal Bati, Professor

Academic Editor

PLOS ONE

Additional Editor Comments (optional):

Dear Author/s,

I would like to thank you for the edits made to this article, which was rejected at the first stage, taking into account the criticisms about the article.
The ethical approach in experimental research, as in this research, is to ensure that the experimental group also benefits from the positive effect after the research.
Long-term correspondence between authors about their articles gives me this confidence.
The issue of the difference in the number of students in the two groups in the study was conveyed as a negative opinion in the previous evaluation, but the authors accepted the criticism and made additional analyzes and presented them in the revised article, showing that the results were similar.
In this context, I think it would be appropriate to publish the article in Plos One after the final evaluation of the reviewers.

Reviewers' comments:

Reviewer's Responses to Questions

**Comments to the Author**

1. If the authors have adequately addressed your comments raised in a previous round of review and you feel that this manuscript is now acceptable for publication, you may indicate that here to bypass the “Comments to the Author” section, enter your conflict of interest statement in the “Confidential to Editor” section, and submit your "Accept" recommendation.

Reviewer #4: All comments have been addressed

Reviewer #5: (No Response)

2. Is the manuscript technically sound, and do the data support the conclusions?

Reviewer #4: Yes

Reviewer #5: Partly

3. Has the statistical analysis been performed appropriately and rigorously? 

Reviewer #4: Yes

Reviewer #5: Yes

4. Have the authors made all data underlying the findings in their manuscript fully available?

Reviewer #4: Yes

Reviewer #5: Yes

5. Is the manuscript presented in an intelligible fashion and written in standard English?

Reviewer #4: Yes

Reviewer #5: Yes

6. Review Comments to the Author

Reviewer #4: Thank you for resubmitting your article for review. I believe that formative OSCEs is a novel educational method that is slowing being adapted by medical schools and is a research area for myself.

Your introduction is clear and sets up your argument for conducting the study.

Your methods (including the blinding process) are clear.

Your results are presented clearly.

You have been clear about your limitations.

The paper reads well and it is very clear to me what you have done within the study.

I hope this paper inspires/ encourages further medical school to implement and utilize formative OSCEs in their curriculum.

I thank you for your hard work in addressing the reviewers comments.

Reviewer #5: I am concerned about the grouping and study design employed in this research. The current findings are based on a sample of 20 students, selected from a larger pool of 1002 students. The formative OSCEs were administered exclusively to this limited group of students. However, I am wondering about the treatment of the control students during this training period. The study's design seems to introduce a bias, as the additional training opportunities provided to the intervention group could inevitably lead to a more favorable training outcome. Additionally, if students in the control group were permitted to view OSCE videos, they might have achieved similar improvements. This casts doubt on the validity of the conclusion on the effectiveness of formative OSCEs.

7. PLOS authors have the option to publish the peer review history of their article (what does this mean?). If published, this will include your full peer review and any attached files.

Reviewer #4: No

Reviewer #5: No

---

## [Editor Report · Acceptance letter]

28 Nov 2023

PONE-D-22-23055R2 

Early formative objective structured clinical examinations for students in the pre-clinical years of medical education: a non-randomized controlled prospective pilot study. 

Dear Dr. Ouldali:

I'm pleased to inform you that your manuscript has been deemed suitable for publication in PLOS ONE. Congratulations! Your manuscript is now with our production department. 

Kind regards, 

on behalf of

Dr. Ayse Hilal Bati 

Academic Editor

PLOS ONE